# A Proposal of a New Nomogram to Predict the Need for Testosterone ReplACEment (TRACE): A Simple Tool for Everyday Clinical Practice

**DOI:** 10.3390/jpm12101654

**Published:** 2022-10-05

**Authors:** Tommaso Cai, Salvatore Privitera, Federica Trovato, Paolo Capogrosso, Federico Dehò, Sebastiano Cimino, Michele Rizzo, Giovanni Liguori, Andrea Salonia, Alessandro Palmieri, Paolo Verze, Truls E. Bjerklund Johansen

**Affiliations:** 1Department of Urology, Santa Chiara Regional Hospital, 38123 Trento, Italy; 2Institute of Clinical Medicine, University of Oslo, 0010 Oslo, Norway; 3Department of Urology, University of Catania, 95100 Catania, Italy; 4Department of Urology, ASST Sette Laghi Hospital, 21100 Varese, Italy; 5Department of Urology, University of Trieste, 34121 Trieste, Italy; 6Department of Urology, University of San Raffaele Vita e Salute, 20019 Milan, Italy; 7Department of Urology, University of Naples, Federico II, 80100 Naples, Italy; 8Department of Medicine, Surgery, Dentistry “Scuola Medica Salernitana”, University of Salerno, 84121 Baronissi, Italy; 9Department of Urology, Oslo University Hospital, 0010 Oslo, Norway; 10Institute of Clinical Medicine, University of Aarhus, 8000 Aarhus, Denmark

**Keywords:** testosterone, androgen deficiency symptoms, LOH, LH, metabolic syndrome

## Abstract

International guidelines suggest to use testosterone therapy (TTh) in hypogonadal men presenting symptoms of testosterone deficiency (TD), even if there is no fixed threshold level of T at which TTh should be started. We aimed to develop and validate a nomogram named TRACE (Testosterone ReplACEment) for predicting the need of TTh in patients with “low–normal” total testosterone levels. The following nomogram variables were used: serum T level; serum LH level; BMI; state of nocturnal erections; metabolic comorbidities; and IPSS total score. The nomogram has been tested by calculating concordance probabilities, as well as assaying the calibration of predicted probability of clinical testosterone deficiency and need for TTh, together with the clinical outcome of the TTh. A cohort of 141 patients was used for the development of the nomogram, while a cohort of 123 patients attending another institution was used to externally validate and calibrate it. Sixty-four patients (45.3%) received TTh. Among them, sixty patients (93.7%) reported a significant clinical improvement after TTh. The nomogram had a concordance index of 0.83 [area under the ROC curve 0.81 (95% CI 0.71–0.83)]. In conclusion, the TRACE nomogram accurately predicted the probability of clinical impairment related to TD, and resulted in a simple and reliable method to use to select hypogonadal patients with not clearly pathological testosterone values who will benefit from TTh.

## 1. Introduction

Although it is now widely recognized that a testosterone deficiency (TD) can negatively impact various organ functions and quality of life (QoL), there is consensus among scientific societies to only offer testosterone therapy (TTh) in symptomatic patients with confirmed low total serum testosterone levels [1,2]. In detail, international guidelines only recommend TTh in the case of compatible signs and symptoms and a subnormal serum testosterone concentration confirmed on two or three separate occasions [1,3]. However, the threshold testosterone value for starting TTh remains controversial [4]. Both the European Association of Urology (EAU) guidelines and the International Consultation for Sexual Medicine (ICSM) suggest starting TTh in the case of hypogonadal symptoms and total testosterone levels of <12 nmol/L (350 ng/dL) [5,6,7], whereas the American Urological Association (AUA) sets as a threshold value for low total testosterone at 10.4 nmol/L (300 ng/dL) [8]. Normal total testosterone values range between 303 to 852 ng/dL (10.5 to 29.5 nmol/L) using the 5th and 95th percentile, respectively [8]. Based on these considerations, at the moment we do not have a clear indication on what the minimum threshold value of total testosterone is at which it is necessary to start the replacement therapy in symptomatic patients. At the same time, male hypogonadism is a cluster of different clinical conditions that require a proper diagnosis before treatment may be considered. Testosterone treatment can reduce symptoms and improve glucose homeostasis, reduce cardiovascular risk and improve life expectancy and QoL [9]. It is therefore important to identify patients with functional hypogonadism and other testosterone deficiency-associated conditions early. Guidelines provide an excellent frame for physicians to individualize treatment, but we consider it desirable to have a fixed threshold level of T, below which symptoms of TD and adverse health outcomes should necessarily indicate TTh.

In order to provide a useful tool for physicians to decide about TTh in everyday clinical practice, we developed and validated a nomogram named TRACE (Testosterone ReplACEment) for predicting the need of TTh in men with symptoms of TD and low-normal total testosterone values.

## 2. Patients and Methods

### 2.1. Study Design and Protocol Schedule

All consecutive patients attending two tertiary institutions for symptoms related to hypogonadism were recruited in this study (testing cohort; Institution 1: Santa Chiara Regional Hospital, validation cohort; Institution 2: Federico II, University of Naples). During the enrollment visit, all demographic, clinical and laboratory parameters that are most frequently associated with TD were collected, and an in-depth urological assessment was also carried out through questionnaires and clinical examination. TTh was initiated by the trialist on the basis of clinical parameters and laboratory results. At 6 months after the initiation of TTh, all patients underwent a follow-up visit, during which the same assessments of the enrollment visit were repeated.

### 2.2. Patients and Data Collection

This study enrolled consecutive hypogonagal patients affected with hypogonadism.

Overall, for the development phase of the tool (phase 1), the data relating to patients enrolled between January 2019 and December 2020 at the Institution 1 were used. For the subsequent validation phase (phase 2), the data relating to patients who were selected on the base of the same selection criteria and enrolled from July 2020 to April 2021 at the Institution 2 were used.

### 2.3. Inclusion and Exclusion Criteria

Male patients were eligible for inclusion in the study if they were older than 18 years, sexually active, presenting with total testosterone values ranging between 12 and 15 nmol/L and associated symptoms related to hypogonadism. We excluded men who already made use of TTh, and/or were receiving 5-ARIs for BPH treatment.

### 2.4. Endocrinological and Clinical Predictors

According to Lunenfeld et al., we considered the following laboratory and clinical predictors for need of TTh: age; serum levels of total T; free T; Sex Hormone Binding Globulin (SBHG); and LH [10]—moreover, according to Kapoor et al., we included Body Mass Index (BMI), metabolic comorbidities (i.e., diabetes, hypertension, hypercholesterolemia or cardiovascular disease), International Index of Erectile Function (IIEF-5) and International Prostate Symptom Score (IPSS) total score [11]. Finally, each male patient was asked to report on the presence of nocturnal erections. Demographic characteristics were collected and recorded during the medical interview by means of standardized questions to obtain homogenous data for analysis. The presence of nocturnal erections was assessed by the patient as “yes” or “no”, and physical activity was registered as regular, seldom or rarely. Patients’ reported outcomes on relief of symptoms after TTh was reported as greatly improved, moderately improved or not improved. The IIEF-5 and IPSS scores were included in the analysis due to the fact that the use of TTh resulted in significant improvement in lower urinary tract symptoms (LUTS) and sexual dysfunction, as reported by Okada [12]. All clinical and laboratory data, as well as questionnaire scores, were used in univariate and multivariate logistic regression models addressing the need of TTh.

### 2.5. Questionnaires and Urological Visits

At the baseline and follow-up evaluation, all patients completed the following questionnaires and tools:-IIEF-5 [13]. The IIEF-5 scale was used to evaluate erectile function in the last six months. The severity of ED was classified severe (IIEF-5 ≤ 7), moderate (IIEF-5 between 8 and 11), mild-moderate (IIEF-5 between 12 and 16), mild (IIEF-5 17–21) and absent (IIEF-5 between 22 and 25).-IPSS [14]. IPSS was used to evaluate lower urinary tract symptoms. A total score of ≤ 7 was considered as normal.-Patient QoL was measured with an Italian version of the SF-36 Health Survey [15]. Higher scores on the SF-36 Health Survey reflect better QoL.

### 2.6. Testosterone Considerations

In line with common clinical practice, serum levels T total, T free, SBHG and LH were measured for all patients at baseline and follow-up examination after 6 months [16,17]. 3 ml of blood serum was taken from all study participants to measure total testosterone and luteinising hormone (LH) [18]. All blood samples were collected between 9 and 11 a.m. from overnight fasting to minimize the circadian rhythm effect on the level of T secretion, in accordance with Crawford [19]. All hormonal assessments were made with chemiluminescent immunoassay methods (Beckman Coulter Inc., Brea, CA, USA). Serum levels of free testosterone (FT) and bioavailable testosterone (Bio-T) were calculated from values of total testosterone, SHBG and serum albumin through the known available formula at http://www.issam.ch/freetesto.htm (accessed on January 2019) [20]. Since there is no precise threshold value of the testosterone level at which to necessarily indicate TTh for the treatment of symptoms of hypogonadism, we have arbitrarily decided to adopt the interval between 12 (349 ng/dL) and 15 nmol/L (432 ng/dL) to start TTh in symptomatic patients, in accordance with the interval identified by Kelleher and Zitzmann [16,17]. Similarly, with regards to the free T level, for which there is not even a precise threshold level to start TTh [3], we decided to adopt 243 pmol/L (70 pg/mL) as the lower limit of the normal range at which to start TTh, in line with Bhasin and the EAU guidelines [2,5,18].

### 2.7. Ethical and Statistical Considerations

All men were informed about the nature of the study, and gave written consent to participate. They were informed that the study would not change the standard management of their clinical condition. In accordance with Italian law, investigators did neither need to request authorization from the institutional review board (IRB) to perform the study, nor collect informed consent from the patients (http://www.agenziafarmaco.gov.it/it/content/linee-guida-studi-osservazionali (accessed on January 2019)). The study was conducted in line with the Good Clinical Practice guidelines and the ethical principles laid down in the latest version of the Declaration of Helsinki. All anamnestic, clinical and laboratory data containing sensitive information about patients were de-identified to ensure analysis of anonymous data only. The de-identification process was performed by non-medical staff by means of dedicated software. Univariate analysis of variance (ANOVA) were performed for all variables and predictors. Differences between groups were evaluated with independent *t* tests for continuous variables and χ^2^ tests for categorical variables. Multivariate analysis of variance (MANOVA) has been used in case of dependent variables. Multivariate relative risk was calculated for all predictors by using Cox proportional hazards regression. Coefficients of multivariate logistic regression models were then used to develop a nomogram predicting the clinical efficacy of TTh (questionnaire and PROs). The variables were selected for the final multivariate model by forward stepwise selection. The accuracy of the score was quantified using the receiver operating characteristics (ROC) curve. To determine the nomogram-predicted probability of clinically significant testosterone deficiency, we applied the score to all 123 men enrolled in the other center during another period (validation set). Accuracy of the score was then quantified using the area under the curve (AUC) for external validation. *p* < 0.05 was considered statistically significant. All reported *p* values were 2-sided. Statistical analyses were performed using SPSS 22.0 for Apple-Macintosh (SPSS, Chicago, IL, USA). All PROs and questionnaire results were included in the evaluation of predictors and compared with the nomogram data.

## 3. Results

Overall, data from 275 consecutive patients (mean age 57.2 ± 9.6) were considered for analysis. Eleven patients were excluded because of missing data, giving a returning rate of 96% (264/275). Data collection was stratified for the two different study phases. Finally, the cohort for phase 1 consisted of 141 patients, while the cohort for phase 2 consisted of 123 patients.

### 3.1. Clinical and Biochemical Findings at Enrollment

Clinical and laboratory characteristics of all patients included in this study are described in Table 1. The mean IIEF-5 score for all patients was 10.8 ± 3.5, while mean IPSS was 8.9 ± 2.1. The mean BMI was 24.2 ± 8.1. Mean total testosterone level resulted at 13.64 ± 1.10 nmol/L, while the LH 9.15 ± 10.3 IU/L.

### 3.2. Testosterone Therapy (TTh)

Sixty-four patients (45.3%) received TTh. Among these, sixty patients (93.7%) reported a significant clinical improvement at the 6-month follow-up visit, as determined in terms of PROs and questionnaire results.

### 3.3. Results of Univariate and Multivariate Analyses

All univariate and multivariate analysis results are shown in Table 2. Among all tested clinical and laboratory predictors, the following parameters have been identified as independent predictors of the need of TTh in men with TD: serum total testosterone level; serum LH level; BMI; presence or absence of nocturnal erections; presence (and number) or absence of metabolic comorbidities; and IPSS score. Serum FT level was associated with the need of TTh in the univariate model (*p* = 0.03), but was not confirmed at the multivariate analysis (*p* = 0.09).

### 3.4. Nomogram Development and Validation

The TRACE nomogram was constructed by incorporating all clinical and laboratory predictors identified by the multivariate analysis (Figure 1).

The concordance index between data obtained from the score and the real data was 0.81 after using bootstrapping to correcting for over-fitting. Calibration plots of the nomogram-predicted probabilities and the improvement in clinical questionnaires and PROs I in the external cohort are displayed in Figure 2.

In the validation data set, the area under the ROC curve was 0.85 (95% CI 0.71–0.93) (Figure 3).

## 4. Discussion

### 4.1. Main Findings

Herein, we developed and validated a nomogram, named TRACE, to help the physician in daily clinical practice to better select patients who need TTh, based on the combination of symptoms related to TD and a total testosterone level ranging between 12 and 15 nmol/L. The clinical and biochemical predictors are easy to collect, and are generally included in the everyday clinical and biochemical patient assessment.

### 4.2. Results in Comparison with Other Studies

The selection of hypogonadal patients who really need and can benefit from TTh is challenging. In case of a total testosterone level below 12 nmol/L, the indication for TTh is clear, and patients experience clinical benefits. This data was recently confirmed by a meta-analysis that showed that TTh only improved libido and erectile function in hypogonadal subjects with significantly reduced T levels (<8 nmol/L and <12 nmol/L, respectively) [21]. In daily clinical practice, however, we often face patients with testosterone levels ranging between normal–low levels and presenting TD symptoms, in whom it is more difficult to establish whether TTh is really necessary and will benefit the patient. Zitzmann et al. reported that TD symptoms may be seen with total testosterone levels as high as 15 nmol/L [17], and patients can benefit from TTh. Behre et al. demonstrated that 6 months of TTh enhanced body composition and quality of life in men aged 50–80 years with hypogonadal symptoms and total testosterone of 515 nmol/L [22]. These men showed further enhancements in body composition and QoL over the following 12 months of TTh. Other authors reported that patients of <40 years of age with total testosterone <400 ng/dL (13.9 nmol/L) had an increase in hypogonadal symptoms after TTh [23]. Indeed, it was this mixture of observations that led to the development and validation of the TRACE nomogram. Many men report unspecific symptoms such as decreased energy and motivation, depressed mood, poor concentration, sleep disturbances and increased body fat [9]. Patients commonly do not pay attention to these symptoms, which could be the first clinical manifestation of androgen deficiency. The appearance of other symptoms of androgen deficiency, such as the lack of nocturnal erections, should lead to T evaluation and consideration of TTh. Recently, Burte C et al. presented the Francophone Society of Sexual Medicine (SFMS) and the French Association of Urology (AFU) recommendations for the management of TD, highlighting disappearance of nocturnal erections as an important symptom of androgen deficiency [24]. On the other hand, a disrupted central nervous system hypothalamo-pituitary network coordination could be the reason for a disturbed neurogenically organized sexual response in aging males [25]. Moreover, TTh has an impact on mood and other androgen deficiency-related symptoms, especially in middle-aged hypogonadal men. We therefore included evaluation of LH levels in our nomogram. In aging men, Leydig cells show structural and functional alterations with an increasingly oxidative intracellular environment and reduced androgen synthesis and serum T levels [25]. Recently, Liang et al. reported that LH has a strong correlation with hypogonadism, suggesting that an increased LH level might be used as an early predictor of forthcoming symptomatic hypogonadism [26]. The same authors showed that patients with higher LH levels also have a higher IPSS score and lower IIEF-5 score, thereby highlighting the role of LH as an independent predictor of worsening IPSS and IEF-5 score. Interestingly, no association had been reported between psychological symptom and total testosterone or LH levels. It is well known that metabolic syndrome and hypogonadism are associated with increased risk of cardiovascular disease [27]. Moreover, some authors recommend TTh in these patients, as long-term therapy with testosterone in men with functional hypogonadism and sexual symptoms reduces obesity parameters, and improves metabolic syndrome and health-related QoL [27,28]. On the basis of these considerations, our nomogram included biochemical, as well as clinical parameters.

### 4.3. Limitations and Strengths of the Present Study

The absence of SHBG and free testosterone levels in the nomogram could be considered a limitation. However, even if these variables are commonly used in everyday clinical practice, our analysis did not demonstrate statistical significance for these as predictors of the need for TTh. In reverse, the inclusion of LH in the nomogram could be regarded as a strength, and is supported by the study of Li et al. which highlighted LH as the strongest factor to identify symptoms related to TD, especially somatic and sexual symptoms [29]. Major strengths are that the development of the nomogram is based on an unselected material of men referred for evaluation of TTh, and the validation of the nomogram in a similar institution. Furthermore, as reported by Black et al., we need to pay attention to the role of androgen receptor polymorphism as a potential contributing factor of the response to TTh [30]. Black et al. then highlighted that neither total testosterone nor bioavailable testosterone had a predictive value for the treatment response [30]. Finally, Tajar et al. demonstrated that symptomatic elderly men with suspected functional hypogonadism should be stratified on the basis of laboratory and clinical features, and compensated hypogonadism should be considered a clinical entity associated with aging. In this sense, our nomogram is able to differentiate between compensated hypogonadism and clinical relevant hypogonadism that requires TTh, due to the fact that both LH and total testosterone and symptoms have been included in the analysis, as suggested by Tajar et al. [31].

## 5. Conclusions

The lack of consensus among guidelines on the threshold level of total testosterone below which the TTh must be indicated highlights the need for additional tools to assist clinicians in defining treatment indications. The TRACE nomogram was developed to meet this need. From our development and external validation work, it results in a simple and reliable method to use to select hypogonadal patients with not clearly pathological testosterone values who will benefit from TTh.

## Figures and Tables

**Figure 1 jpm-12-01654-f001:**
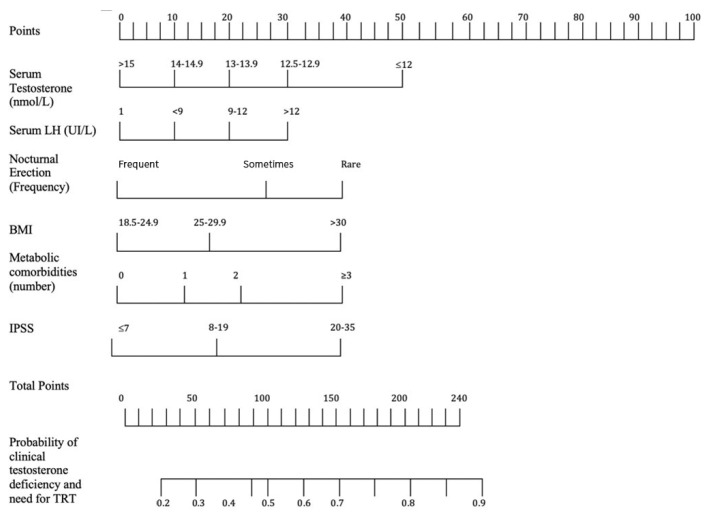
The figure shows the TRACE nomogram. In order to use the nomogram, draw a line perpendicular from the corresponding axis of each risk factor until it reaches the top line labeled “Points.” Sum up the number of points for all risk factors then draw a line descending from the axis labeled “Total Points” until it intercepts each of the survival axes to determine the probability of clinical testosterone deficiency and the need for testosterone therapy.

**Figure 2 jpm-12-01654-f002:**
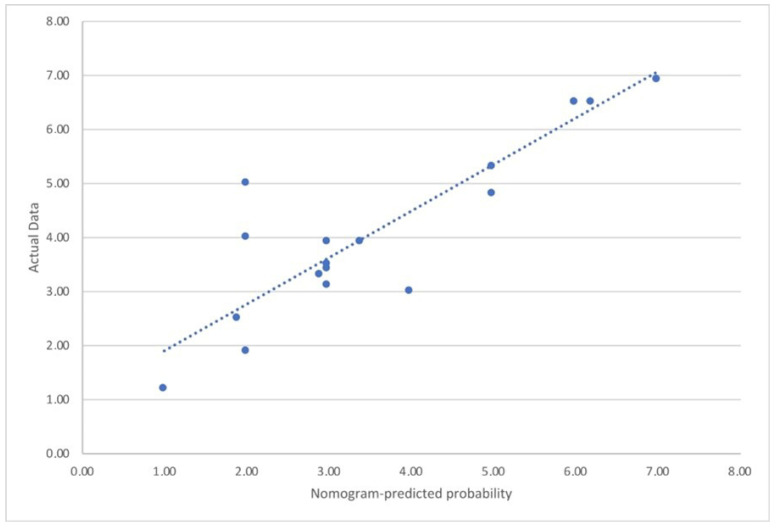
The figure shows the calibration curves for the nomogram-predicted probability of testosterone deficiency and the need for testosterone therapy.

**Figure 3 jpm-12-01654-f003:**
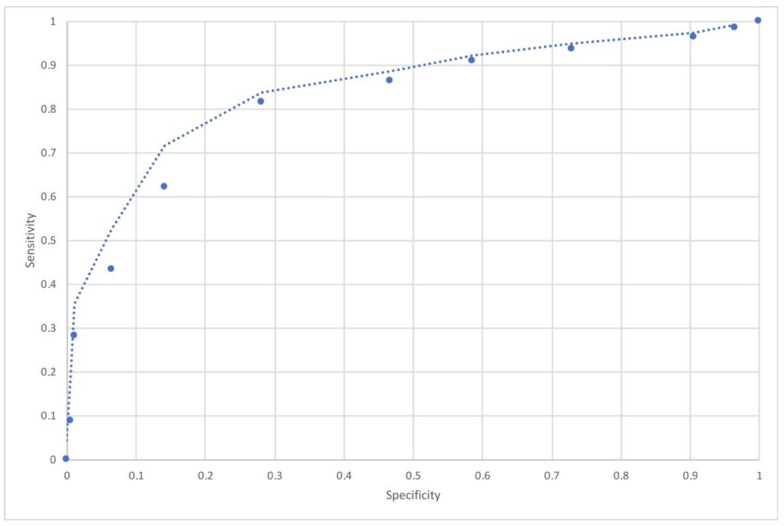
The figure shows the ROC curve analysis.

**Table 1 jpm-12-01654-t001:** Patient clinical and laboratory characteristics at enrolment time, stratified by study phase.

Total Analyzed Patients	264		
			*p*
Group	Training cohort	Validation cohort	
No. of patients	141	123	
Mean age (±SD^†^)	56.9 ± 7.5	57.4 ± 9.5	0.63
Body Mass Index (mean ± SD^†^)	23.9 ± 9.2	24.5 ± 7.5	0.56
Waist circumference (cm)	87 ± 5.3	86 ± 6.1	0.15
Charlson Comorbidity Index (median; range)	2 (1–3)	2 (1–3)	-
Baseline total testosterone (nmol/L) (mean ± SD^†^)	13.8 ± 1.24	13.1 ± 1.01	0.77
Baseline free-testosterone (ng/dL) (mean ± SD^†^)	4.1 ± 2.13	3.9 ± 2.6	0.49
LH* (IU/L) (mean ± SD^†^)	9.09 ± 10.4	9.18 ± 10.8	0.94
FSH§ (IU/L) (mean ± SD^†^)	7.4 ± 1.1	7.2 ± 1.3	0.17
SHBG° (nmol/L) (mean ± SD^†^)	43.9 ± 25.7	49.5 ± 33.1	0.12
IIEF-5# (mean ± SD^†^)	10.5 ± 4.3	11.0 ± 4.1	0.33
IPSS$ (mean ± SD^†^)	8.6 ± 2.0	9.1 ± 3.2	0.12
PSA+ (mean ± SD^†^)	3.3 ± 1.9	3.6 ± 2.3	0.24
Total serum glucose (mg/dL) (mean ± SD^†^)	97 ± 17.2	93 ± 19.8	0.08
Total serum cholesterol (mg/dL) (mean ± SD^†^)	198 ± 23.1	201 ± 25.1	0.31

The table shows all patient clinical and laboratory characteristics at enrolment time. SD^†^ = standard deviation; LH* = Luteinising Hormone; FSH§ = Follicle Stimulating Hormone; SHBG° = Sexual Hormone Binding Protein; IIEF-5# = International Index of Erectile Function; IPSS$ = International Prostate Symptom Score; and PSA+ = Prostate Specific Antigen.

**Table 2 jpm-12-01654-t002:** Univariate and multivariate analysis results of laboratory and clinical factors affecting testosterone replace treatment need in 141 patients enrolled in the training set.

Categories (Variables)	Univariate Analysis (p) HR* (95% CI^†^)	Multivariate Analysis (*p*) HR* (95% CI^†^)
Age	0.76 (HR 0.81; 95% 0.17–1.22)	0.56 (HR 0.98; 95% 0.23–1.10)
Body Mass Index	0.03 (HR 2.3; 95% 1.41–3.31)	0.003 (HR 1.97; 95% 1.56–2.25)
Nocturnal Erection	0.01 (HR 3.8; 95% 2.57–4.98)	0.001 (HR 4.01; 95% 2.63–5.80)
Waist circumference	0.09 (HR 1.22; 95% 0.67–1.98)	0.07 (HR 1.10; 95% 0.64–1.77)
Charlson Comorbidity Index	0.01 (HR 3.06; 95% 2.00–3.99)	0.003 (HR 2.97; 95% 1.50–3.67)
Serum total testosterone	0.02 (HR 5.13; 95% 3.16–6.18)	0.001 (HR 5.60; 95% 4.87–6.81)
Serums free–testosterone	0.03 (HR 1.02; 95% 0.79–1.23)	0.09 (HR 1.97; 95% 1.11–3.33)
LH*	0.04 (HR 2.98; 95% 1.56–3.42)	0.03 (HR 3.44; 95% 2.80–3.77)
FSH^§^	0.75 (HR 1.03; 95% 0.43–1.12)	0.65 (HR 0.97; 95% 0.60–1.06)
SHBG°	0.01 (HR 0.91; 95% 0.17–1.94)	0.09 (HR 0.71; 95% 0.07–1.01)
IIEF-5#	0.03 (HR 2.16; 95% 1.98–2.77)	0.003 (HR 3.17; 95% 2.54–3.88)
IPSS$	0.02 (HR 2.96; 95% 2.11–4.07)	0.001 (HR 3.44; 95% 2.81–5.89)
PSA°	0.34 (HR 0.56; 95% 0.09–0.78)	0.65 (HR 0.91; 95% 0.66–1.02)

The table shows univariate and multivariate analysis results of laboratory and clinical factors affecting testosterone/replace treatment need in 141 patients enrolled in the training set. SD^†^ = standard deviation; LH* = Luteinising Hormone; FSH^§^ = Follicle Stimulating Hormone; SHBG° = Sexual Hormone Binding Protein; IIEF-5# = International Index of Erectile Function; IPSS$ = International Prostate Symptom Score; and PSA° = Prostate Specific Antigen.

## Data Availability

Not applicable.

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
