# Peer review of "A Proposal of a New Nomogram to Predict the Need for Testosterone ReplACEment (TRACE): A Simple Tool for Everyday Clinical Practice"

_jpm, 2022, doi:10.3390/jpm12101654_

Round 1

Reviewer 1 Report

A tool to be used in clinical practice to decide whether a patient with "low-normal" testosterone and symptoms of hypogonadism would benefit from testosterone therapy would provide a major advance. Until now, guidelines suggest an empirical trial for a limited period of time in such patients. A nomogram could largely improve such an approach and, if validated properly, be integrated into future guidelines.

Comments in detail:

1. For practitioners unfamiliar with the use of nomograms, there should be detailed instructions. Examples should be provided for patients who do or do not qualify for testosterone therapy. After all, this nomogram is intended for everyday practice, and the practitioner should not be left alone to figure out how to use this novel tool.

2. The title of the paper should clearly indicate that the TRACE nomogram has been developed for patients with "low-normal" total testosterone levels between 12 and 15 nmol/L. The current title suggests that this tool could be used for any candidate for testosterone therapy.     

3. Please consider replacing the term "testosterone replacement therapy" by "testosterone therapy". Not only has this been suggested in ref. 7 and elsewhere, but especially in men who are considered in the - arbitrarily assumed - "normal" range, the term "replacement" appears inadequate. It may apply in cases of unequivocally low testosterone but not in men who, according to most guidelines, have testosterone in the "normal" range.

4. Please note that "LOH" is also considered by many experts as a term that should no longer be used. The term "functional hypogonadism" has been suggested. This should be discussed.

5. Page 2, first line: "International" (not "Internal"). Please correct. (This may not be shown by a spell check.)

6. Please check all references. Some numbers are incorrectly allocated.

7. Table 1: It seems quite unusual that the BMI of the cohorts is in the normal weight range. Most hypogonadal patients are either overweight or obese, and obesity affects SHBG levels. One could question whether the patients used for developing the nomogram represent the typical patients presenting with hypogonadism.

8. Discussion: The statement that "the indication for testosterone therapy in patients with total testosterone levels <12 nmol/L is clear" is not at all straightforward. Not only do several guidelines (Endocrine Society, AUA) recommend a threshold of 10.4 nmol/L and the EAA consider total testosterone between 8 and 12 nmol/L as "borderline", there also is limited information as to whether patients with total testosterone <12 nmol/L respond to testosterone therapy. This was demonstrated in an old paper by Morales' group in Canada (Black et al. BJU Int 2004;94:1066-1070). - Please discuss androgen receptor polymorphism as a potential contributing factor.

9. Discussion: Since LH is one of the criteria used for the nomogram, the concept of compensated hypogonadism as described by Tajar et al. for the EMAS study (J Clin Endocrinol Metab 2010;95:1810 –1818) should be discussed.

10. Discussion, page 9, line 4: Please rephrase "patients don't put attention" (suggestion: patients do not pay attention).

Author Response

Comments:

A tool to be used in clinical practice to decide whether a patient with "low-normal" testosterone and symptoms of hypogonadism would benefit from testosterone therapy would provide a major advance. Until now, guidelines suggest an empirical trial for a limited period of time in such patients. A nomogram could largely improve such an approach and, if validated properly, be integrated into future guidelines.

Comments in detail:

  1. For practitioners unfamiliar with the use of nomograms, there should be detailed instructions. Examples should be provided for patients who do or do not qualify for testosterone therapy. After all, this nomogram is intended for everyday practice, and the practitioner should not be left alone to figure out how to use this novel tool.

Response: In line with your suggestion, the following sentence has been added in the figure 1 legend: In order to use the nomogram, draw a line perpendicular from the corresponding axis of each risk factor until it reaches the top line labeled "Points." Sum up the number of points for all risk factors then draw a line descending from the axis labeled "Total Points" until it intercepts each of the survival axes to determine the probability of clinical testosterone deficiency and the need for testosterone therapy.

  1. The title of the paper should clearly indicate that the TRACE nomogram has been developed for patients with "low-normal" total testosterone levels between 12 and 15 nmol/L. The current title suggests that this tool could be used for any candidate for testosterone therapy.

Response: In line with your suggestion, the following sentence has been added in the abstract and in the text: in patients with "low-normal" total testosterone levels. Many thanks for your suggestion.

  1. Please consider replacing the term "testosterone replacement therapy" by "testosterone therapy". Not only has this been suggested in ref. 7 and elsewhere, but especially in men who are considered in the - arbitrarily assumed - "normal" range, the term "replacement" appears inadequate. It may apply in cases of unequivocally low testosterone but not in men who, according to most guidelines, have testosterone in the "normal" range.

Response: In line with your suggestion, the phrase "testosterone replacement therapy" has been replaced by the following: testosterone therapy Many thanks for your suggestion.

  1. Please note that "LOH" is also considered by many experts as a term that should no longer be used. The term "functional hypogonadism" has been suggested. This should be discussed.

Response: In line with your suggestion the terms LOH has been replaced by the functional hypogonadism.

  1. Page 2, first line: "International" (not "Internal"). Please correct. (This may not be shown by a spell check.)

Response: Many thanks for your suggestion.

  1. Please check all references. Some numbers are incorrectly allocated.

Response: All references have been revised and checked.

  1. Table 1: It seems quite unusual that the BMI of the cohorts is in the normal weight range. Most hypogonadal patients are either overweight or obese, and obesity affects SHBG levels. One could question whether the patients used for developing the nomogram represent the typical patients presenting with hypogonadism.

Response: Many thanks for your suggestion. The table has been revised and checked in line with your comments.

  1. Discussion: The statement that "the indication for testosterone therapy in patients with total testosterone levels <12 nmol/L is clear" is not at all straightforward. Not only do several guidelines (Endocrine Society, AUA) recommend a threshold of 10.4 nmol/L and the EAA consider total testosterone between 8 and 12 nmol/L as "borderline", there also is limited information as to whether patients with total testosterone <12 nmol/L respond to testosterone therapy. This was demonstrated in an old paper by Morales' group in Canada (Black et al. BJU Int 2004;94:1066-1070). - Please discuss androgen receptor polymorphism as a potential contributing factor.

Response: In line with your suggestion, the discussion has been improved and the suggested reference has been added to the reference list.

  1. Discussion: Since LH is one of the criteria used for the nomogram, the concept of compensated hypogonadism as described by Tajar et al. for the EMAS study (J Clin Endocrinol Metab 2010;95:1810 –1818) should be discussed.

Response: In line with your suggestion, the discussion has been improved and the suggested reference has been added to the reference list.

  1. Discussion, page 9, line 4: Please rephrase "patients don't put attention" (suggestion: patients do not pay attention).

Response: Many thanks for this suggestion.

Reviewer 2 Report

Ø  The authors report a very interesting, and timely examination of nomogram named TRACE (Testosterone Replacement) for predicting the need of TRT and highlighting its efficacy by the probability of clinical testosterone deficiency and proved to identify patients with significant benefit of testosterone replacement. 

Ø  The work is not well-illustrated to attract high interest with the Journal of personal medicine readership. Unfortunately, the whole introduction and result discussion seems to be vague and quite confusing. So, I request authors to look carefully and modify the manuscript with native English speaker.  

Ø  As such, I recommend acceptance after first considering full modification of manuscript with English Editing.

Author Response

Comments:

The authors report a very interesting, and timely examination of nomogram named TRACE (Testosterone Replacement) for predicting the need of TRT and highlighting its efficacy by the probability of clinical testosterone deficiency and proved to identify patients with significant benefit of testosterone replacement.

The work is not well-illustrated to attract high interest with the Journal of personal medicine readership. Unfortunately, the whole introduction and result discussion seems to be vague and quite confusing. So, I request authors to look carefully and modify the manuscript with native English speaker.

Response: Introduction and discussion have been revised.

As such, I recommend acceptance after first considering full modification of manuscript with English Editing. 

Response: The full manuscript has been revised by a native English speaker.

Round 2

Reviewer 1 Report

The manuscript has been largely improved and can be accepted with one important and one minor correction.

Important change: Please change the acronym for testosterone therapy to TTh. TT is often used for "total testosterone". TTh is also used in reference 7.

Minor correction: Page 5, line 2: "for analysis" (not: fro analaysis).

Author Response

Response to the Editor

Dear Editor,

I’m submitting a revised version of the manuscript (JPM-1922213), in line with the Editor’s and Referee’s comments. Thank you for your attention to our paper.

Reviewer #1 (Comments to the Author):

The manuscript has been largely improved and can be accepted with one important and one minor correction.

Important change: Please change the acronym for testosterone therapy to TTh. TT is often used for "total testosterone". TTh is also used in reference 7.

Minor correction: Page 5, line 2: "for analysis" (not: fro analaysis).

Response: In line with your suggestion, the acronym TT has been replace d by the TTh. The phrase fro analaysis has been replaced by the following: for analaysis

Many thanks for your suggestion.

Reviewer 2 Report

I appreciate authors efforts and Now manuscript looks appropriate to be published in JPM.

Author Response

Response to the Editor

Dear Editor,

I’m submitting a revised version of the manuscript (JPM-1922213), in line with the Editor’s and Referee’s comments. Thank you for your attention to our paper.

Reviewer #2 (Comments to the Author):

I appreciate authors efforts and Now manuscript looks appropriate to be published in JPM.

Response: Many thanks for your comment.